# ‘This Is What the Colour Green Smells Like!’: Urban Forest Bathing Improved Adolescent Nature Connection and Wellbeing

**DOI:** 10.3390/ijerph192315594

**Published:** 2022-11-24

**Authors:** Kirsten McEwan, Vanessa Potter, Yasuhiro Kotera, Jessica Eve Jackson, Sarah Greaves

**Affiliations:** 1College of Health, Psychology and Social Care, University of Derby, Kedleston Road, Derby DE22 1GB, UK; 2ParkBathe, London SE19 2RP, UK; 3School of Health Sciences, University of Nottingham, Queen’s Medical Centre, Nottingham NG7 2HA, UK; 4Support and Wellbeing Service (Student and Campus Life), University of Nottingham, University Park, Nottingham NG7 2RD, UK

**Keywords:** adolescent, anxiety, forest bathing, nature connection, rumination, social connection

## Abstract

Background: Research suggests that an early connection with nature can benefit wellbeing into adulthood. However, there is less research assessing whether adolescents benefit from formal nature connection interventions such as forest bathing (slow mindful nature walks). This research aimed to assess whether an urban nature connection intervention (called ParkBathe) could improve adolescents’ nature connection and wellbeing. Method: In an experimental repeated measures design, 44 adolescents sampled opportunistically from Scouts groups, completed surveys and interviews before and after experiencing an urban nature connection intervention. Results: Paired-samples *t*-tests between baseline and post-intervention survey scores revealed statistically significant improvements in anxiety (13% reduction); rumination (44% reduction); scepticism (17% reduction); nature connection (25% increase); and social connection (12% increase). The largest effect size was found for nature connection. Interviews revealed that before the session, participants had a mixed understanding and expectations of the intervention. Conclusions: After the session, the participants expressed enjoying the social aspects of being part of a group and being present in the moment by noticing nature. They expressed the effects of this as immediately calming and relaxing. Urban forest bathing improved nature connection and wellbeing in adolescents and could be implemented and/or signposted by schools and youth charities.

## 1. Introduction

An estimated 20% of adolescents struggle with poor mental health [1], with rates of anxiety, depression, and loneliness being amplified during the pandemic [2,3]. Some researchers argue that a combination of screen addiction and a lack of time outside might be contributing factors to poor mental wellbeing [4]. The shortage in resources for young people’s mental health means that the issue has shifted to society to address [5], and increasingly, organisations such as Natural England are exploring how spending time in nature can help young people’s wellbeing [6]. Some research has indicated that if people do not have a nature connection in childhood, they are more likely to remain disconnected as an adult [7]. In addition, around age 11, many young people experience a reduction in nature connection, famously termed ‘Nature Deficit Disorder’ [8], and may not rediscover their connection until age 30 [9]. Some might argue, therefore, that encouraging a nature connection in young people might offer a buffer against poor mental wellbeing in adolescence through to adulthood [10].

There are a wealth of correlational studies showing that mere access (i.e., without intervention) to nature is beneficial for adolescents’ wellbeing. A review by McCormick [11] found that access to nature was associated with improved wellbeing, health, social relationships, stress, cognitive development, attention restoration, memory, self-discipline, behaviours and symptoms of ADHD, and standardised test scores. In addition to studies correlating access to nature with wellbeing, surveys have found that more access to nature was associated with fewer mental health issues [12]; improved ability to cope with adversity [13], improved ability to cope with problems, ability to think clearly, feeling relaxed [14], improved behaviour [15], improved cognition, concentration, health, and wellbeing [16,17]. In an experimental study, Li et al. [18] used GPS trackers and surveys to monitor the effect of nature access on mood. They found that the average time adolescents spent outdoors was 50 min per day, and greater access and time spent outdoors was associated with better mood. Following the pandemic, 70% of adolescents said that they would like to spend more time outdoors with friends [19].

Beyond correlational studies exploring the benefits of merely accessing nature, there have been some experimental studies assessing interventions that aim to improve adolescents’ nature connection (i.e., their feelings of connection with nature) and wellbeing [6,20,21,22]. Most of the nature connection activities described in previous studies appear to be forest-school- or forest-bathing-inspired. According to the Forest School Association, forest school is a child-centred learning process that supports play, exploration, and risk taking through practical experiences in a natural setting. Forest bathing is a slow, mindful nature walk where participants are guided to pay close attention to their surroundings using their senses. Where applied to children, forest bathing appears to take more diverse forms, and might encompass playful and creative sensory activities. In a review of 17 papers describing forest bathing in primary-aged children, Song and Bang [23] found that a range of activities (e.g., forest play, engaging with the five senses, meditation, walking, and observing animals) improved depression, anxiety, stress, anger, self-esteem, prosociality, and school adjustment. Hohashi and Kobayashi [24] found that two trips comprising walking in the forest for 30 min and sketching the forest for 15 min improved salivary amylase (an enzyme associated with stress), relaxation, and negative mood in adolescent girls, compared with conducting the same activities in the city.

Whilst brief nature connection interventions (e.g., [24]) have been beneficial, several studies have examined the benefits of longer-term interventions. For example, Bang et al. [25] found a significant improvement in self-esteem and a decrease in depressive symptoms following 10 sessions of play activities and a session on the five senses in an urban forest. Chang et al. [26] found improvements in anxiety, depression, and self-concept following four sessions of woodland play, orienteering, crafts, and drama. Barton et al. [27] found that wilderness expeditions, lasting an average of 5 days, improved self-esteem and nature connection in adolescents (especially girls).

Nature connection interventions have also been shown to be helpful for adolescents with challenging behaviours. For example, Jeon et al. [28] found that teenagers on probation showed improved wellbeing and heart rate variability (an indicator of good heart health and relaxation) after 2 days of forest bathing and creative activities. Whilst Machácková et al. [29] found that 16 sessions of observing prosocial behaviours of animals and insects improved teenage delinquents’ scores on psychopathology, irritability, egocentricity, emotional instability, restlessness, and negativism. 

These formal nature connection intervention studies provide promising evidence for the potential to improve adolescent nature connection and wellbeing. However, most previous studies have assessed mere access to nature, relying on correlational survey data, and have not used experimental designs that involve assessing outcomes before and after delivering a more active nature connection intervention. The current study is therefore unique in using a mixed methods experimental design, collecting survey and interview data before and after a formal nature connection intervention. In this study, forest bathing was delivered to adolescents, and accordingly, was adapted to suit the needs of younger people by encompassing more playful and creative sensory activities, such as creating artwork from natural materials and interviewing a tree about its life (more details can be found in the methods section). Whilst most previous intervention studies conducted forest bathing interventions in wilderness or rural settings, young people might be limited in their access to such wilderness or rural settings due to their lack of autonomy, availability of transport, and parental permission [4]. The current study is unique in the respect that the intervention is situated in more accessible urban parks and encourages adolescents to connect with nearby nature, with the goals of increasing feasibility of access and ease of continued practice.

### Aim

This research aimed to assess whether an urban nature connection activity (called ParkBathe) could improve adolescents’ nature connection and wellbeing, by evaluating their experiences before and after the intervention through surveys and interviews. We hypothesise that nature connection and wellbeing survey scores will improve following a formal nature connection activity that is based on forest bathing.

## 2. Method

### 2.1. Design

The evaluation used a mixed methods repeated measures design, where participants provided survey and interview data before and after a 1.5 h session of urban forest Bathing. Through the repeated measures design, participants acted as their own controls, and their baseline and post-intervention data were matched through a participant-generated ID code.

Survey and interview data were collected immediately before and after a 1.5 h urban forest bathing session (called ParkBathe). Transcripts from interviews were thematically analysed to capture adolescents’ more in-depth experience of ParkBathe, using this feedback to improve future sessions and make it accessible to a wider audience.

### 2.2. Participants

The study was approved by the College of Health, Psychology and Social Care, University of Derby Research Ethics Committee, and was performed in accordance with the ethical standards as laid down in the 1964 Declaration of Helsinki. The intervention was delivered during the COVID-19 pandemic and aimed to improve mental wellbeing and social connection in adolescents who had been unable to attend school or meet with friends. Adolescents were sampled opportunistically from Scouts groups. The authors contacted Scout leaders local to the intervention site and asked them to invite their Scout groups and obtain parental permissions and consent. Scout leaders attended the sessions with their Scout groups to provide a sense of safeness through familiarity, and to comply with safeguarding requirements. Seventy-seven adolescents were registered by their parents to attend the session; of these, 63 adolescents completed baseline surveys, 51 completed post-intervention surveys, and 44 adolescents were matched according to their ID code. Hence, after an initial loss of complete data following registration (43% loss), the final sample completing the pre–post assessments was 44 adolescents (17 females (44%), 19 males (49%), and 3 other or prefer not to say (8%); age 13.05 ± 1.65, range 9–17 years). There were 30 white participants (77%), 4 of mixed ethnicity (10%), 3 Asian participants (8%), and 2 ‘others’ (5%). Nineteen adolescents took part in field-based interviews immediately before and after the ParkBathe session. The researchers did attempt to collect follow-up data at 3 months but had a low response rate (*N* = 4), and felt that the data could not be considered generalisable.

### 2.3. Procedure

The practitioners approached Scout and Youth group leaders who were local to the intervention site of Crystal Palace Park, London, and invited them to book their groups of adolescents onto a ParkBathe session. The group leaders distributed the information sheet and consent form via email to the adolescents’ parents to complete on their child’s behalf. If adolescents were aged 16–18 years old, they completed their own consent forms. On arrival at their session, the participants were asked to complete an online or paper-based survey and were interviewed. They began the ParkBathe session, which comprised a 1.5 h guided walk led by two qualified forest bathing guides around an urban park (Crystal Palace Park, London, UK). If participants did not wish to consent to the evaluation (*N* = 5), they could still engage in ParkBathe.

### 2.4. Outcome Measures

#### 2.4.1. Survey

The online survey comprised a total of 11 items with Likert responses. Measures included anxiety (tension subscale of the profiles of mood states (POMS), six items scored 1–5 [30]), rumination (one item scored 1–7 [31]), social connection (inclusion of other in the self scale, one item scored 1–7 [32]), nature connection (inclusion of nature in self scale, one item scored 1–7 [33]), scepticism about forest bathing (one item scored 1–7, created by the researchers for this study), and current mood as measured by Gilbert’s three circle model of affect [34] (one item created by the researchers for this study, which measures whether the participant predominantly has feelings of calm, excitement, or anxiety). 

#### 2.4.2. Interview

Adolescents were asked the following questions before their ParkBathe session: Have you ever done something like forest bathing before? What do you think this is about? How are you feeling about it? How do you think it will make you feel? How is school at the moment? The following questions were asked after ParkBathe: How did you find that? Was it what you were expecting? How do you feel now? Did you have a favourite exercise? Do you think you might come and use the park to relax more? Adolescents were interviewed and audio was recorded using a mobile phone with plug-in microphone. Audio was transcribed and then thematically analysed by an independent researcher (a qualitative researcher with a background in child and adolescent nursing), and the analysis was second-coded, discussed, and agreed with a second independent researcher (a qualitative researcher with a PGDip in youth and community development). Independent researchers conducted the transcription and analysis because the research team wanted to avoid potential bias that might arise from team members who delivered the intervention conducting the transcription and analysis. Soundbites from adolescent interviews and further discussions about the adolescent sessions can be found on the ParkBathe podcast https://podcasts.apple.com/gb/podcast/mother-nature-connecting-children-to-nature/id1543205446?i=1000548655401 (accessed 23 August 2022).

#### 2.4.3. Intervention

The free 1.5 h forest bathing sessions were led by two qualified practitioners trained by The Forest Bathing Institute (Surrey, UK), and involved the guided discovery and mindful appreciation of an urban park (Crystal Palace Park, London, UK). Forest bathing group sizes ranged from 6–15 participants. The practitioners originally tried to deliver a similar forest bathing session to the type that adults would receive (e.g., walking in silence whilst engaging in structured sensory activities). They quickly realised from observations that the adolescents were too active and talkative for this to be enjoyable for them, and so sessions were iteratively improved using adolescents’ feedback. The sessions therefore resembled forest school, but with a more sensory forest bathing focus. 

The session started with an introduction in which the history and purpose of forest bathing was explained. Participants engaged in visual activities which included (i) visually inspecting a tree from roots to canopy, and noticing the shape and character; (ii) searching for leaves and other fallen objects to create a natural colour palette (perhaps colours they would like to paint their bedroom); and (iii) inspired by images of art produced by Andy Goldsworthy, to create their own natural artwork on the forest floor. In a listening activity, adolescents were invited to cup their ears to determine what sounds they could hear. In a smell activity, adolescents were invited to smell leaf litter and soil from the woodland floor; however, this activity was observed to make them uncomfortable and self-conscious, and was therefore omitted from future sessions. In a touch activity, they were invited to partner-up, and one person guided the other (who had their eyes closed) to a tree and encouraged them to get to know their tree through touch alone, they then brought their partner back to the centre and asked them to find their tree. Finally, in an empathy activity, adolescents interviewed a tree in pairs, with one person pretending to be the tree whilst another interviewed the tree about their life. 

Adolescents also engaged in two sharing circles throughout the session, where they were invited to feedback on their experience. The intention of sharing circles is peer learning and benefiting from the experience of others [35]. For example, one participant may notice a unique colour of leaf, causing other participants to seek out the same leaf.

## 3. Results

Three difference scores in anxiety (POMS), one in rumination, and two in scepticism between pre-intervention and post-intervention sessions were identified as outliers by the outlier labelling rule [36], and were therefore Winzorised [37]. The difference scores for all dependent variables from pre-intervention to post-intervention were normally distributed, as assessed by Q–Q plots.

Paired-samples *t*-tests were conducted for all outcome variables where complete data was provided between baseline and post-intervention (*N* = 44). There were statistically significant improvements in nature connection and social connection, which both increased significantly, and improvements in anxiety (POMS), rumination, scepticism, and excitement (three circle model), which reduced significantly (Table 1). The effect size was largest for nature connection (*d* = 0.95 [38]). 

### 3.1. Subgroup Analyses

Subgroup analyses were conducted to examine whether there was a significant interaction effect between subgroups from baseline to post-intervention: gender (female vs. male), nature connection, scepticism, and anxiety at baseline (high vs. low scores were calculated based on the median). Two-way mixed ANOVAs were conducted. There was homogeneity of variances (Levene’s test of homogeneity of variance, *p* > 0.05) and covariances (Box’s test of equality of covariance matrices, *p* > 0.001). For gender (female vs. male), significant interaction effects were identified in anxiety (POMS) and scepticism (see Figure 1). Female adolescents’ anxiety decreased substantially (baseline 1.94 ± 0.15, post 1.53 ± 0.15) relative to male adolescents’ change, which was a subtle increase (baseline 1.51 ± 0.14, post 1.54 ± 0.14) (*F*(1, 34) = 4.83, *p* = 0.04, partial η^2^ = 0.12).

Likewise, as shown in Figure 2, females’ scepticism decreased substantially (Pre 3.47 ± 0.44, Post 2.41 ± 0.46) relative to males’ change (Pre 2.90 ± 0.42, Post 2.90 ± 0.43) (*F*(1, 34) = 5.61, *p* = 0.02, partial η^2^ = 0.14).

As shown in Figure 3, the low nature connection group at baseline increased in nature connection substantially (pre 3.14 ± 0.16, post 4.93 ± 0.23) relative to the high nature connection group (pre 5.64 ± 0.26, post 5.64 ± 0.37) (*F*(1, 37) = 20.84, *p* < 0.001, partial η^2^ = 0.36). 

### 3.2. Qualitative Findings

The qualitative findings support these survey results. Firstly, there was a mix of understanding and expectations from the adolescents. There were participants who did not have any understanding of forest bathing at all. For example, one participant stated, *p8* ‘*Never heard of it until now, it’s a new thing’*, implying that they thought it was a contemporary form of practice. Other participants related it to more established wellbeing practices such as mediation, yoga, and mindfulness. However, most of the participants recognised that it was about being in nature. For example, one participant stated, *p2 ‘It’s about the nature around us that seems to be doing something to you*’, another stated, *p5* ‘*Immersion in nature*’. This indicates that there was a collective understanding that a key element of this approach involved the therapeutic factors of being in the natural world. 

Five participants commented on their current mood before taking part; again, this ranged from feeling relaxed to stressed. For example, one participant stated, *p1* ‘*I’m quite relaxed*’ but another stated *p7* ‘The *only way to be calm is if anger was not an emotion’*. There were some expectations that it would improve their mental health. For example, *p6* ‘*It’s an opportunity to relax without having to think’*. However, a parent of a child admitted *‘He didn’t come voluntarily but I know he’ll enjoy it when he gets going’.* This implies that this parent recognised that their child was reluctant but believed that taking part would be beneficial. 

The qualitative findings following the intervention indicate that for most of the participants the experience was seen as positive, and this was expressed in several ways. Firstly, there was a strong sense of enjoyment. For example, one participant said, *p4* ‘*It was fun, creative’* and another *p9 ‘To explore, do activities, enjoy being here’*. One participant stated, *p12 ‘Everyone here is much younger, and I just got to be a kid again’.* This indicates that this young person was able to be childlike and carefree. Secondly, the social aspect of the activities was also something some participants particularly noticed. For example, one participant stated, *p6 ‘It took trust to actually listen to people’.* This implies that there was a sense in building confidence as a collective to connect with one another. Another participant suggested they were surprised there was a focus on group interaction *p13* ‘*I thought it would be more about nature and me, but it was less about that and more about the connection with activities we did’.*


There was strong indication from the participants that they were able to be present within the moment by noticing the natural world around them. For example, one participant stated, *p14 ‘Making yourself stop, take stock and take time to listen is great’.* Other participants expressed what they had noticed whilst doing the activities. For example, *p10 ‘(The tree bark) Smells like fresh bread, the doughiness of the bread’* and *p3 ‘Trees grow in very peculiar ways, none of them are ever the same’.* Other participants conveyed learning from the experience. For example, *p7 ‘I never realised how brittle these trees are. You think they’ll be hard because they’ve been here so long, but you touch it and the bark flakes off. Imagine how long these trees have been here and how much knowledge they’ve got; they’ve seen so much.’* and *p1 ‘Having a conversation with a tree was enlightening’.* This connection to nature was described further by another participant who stated, *p6 ‘When I feel the tree, you have a connection with the tree, how it’s feeling (not like human emotions), like the health of the tree. It’s like I’m looking for a heartbeat, and as you move your fingers down the tree, it’s almost like you can feel it’.* Here the participant associated the tree as a living entity they could relate to. Another participant’s feeling of connection was described as rooted historically, *p12 ‘Trees are beautiful, they keep us alive, they do so much for us, we can misjudge nature but when you have that connection with nature it is like that instant bond. Nature, it is almost like part of who we are and our ancestors and things and connecting to deeper history’.*

Lastly, there was an overall recognition that the activities had a positive impact on their mental health. Participants described feeling *p18* ‘*Much more relaxed and rested*’, *p8 ‘Found activities calming’, p2 ‘Relaxing experience’* and *p4 ‘Very chill’.* Another participant expanded on this, *p9* ‘*Much happier. I’ve had a lot of stress this week. Exams coming up, but I’m not even thinking about that right now, I’ve got chronic fatigue so my fight/flight is all over the place, next time I’ll spend more time in the woods, just listening and jumping about.*’ One participant expressed a deeper value in the activity they experienced, *p7 ‘I found this experience to be incredibly indulgent and an eye-opening journey I truly hope this reaches other kids as I believe it helps and can bring enlightenment to those without’.* For this participant, taking part led to an immediate revelation and insight into how this type of practice can be beneficial. Therefore, in summary, these findings indicate an immediate effect on connecting adolescents with people, nature, and themselves.

## 4. Discussion

### 4.1. Survey Results

When interpreting the below findings, it is worth considering the context that the intervention was delivered during the COVID-19 pandemic, and aimed to improve mental wellbeing and social connection in adolescents who had been unable to attend school or meet with friends. There were statistically significant improvements in nature connection and social connection, which both increased significantly, and significant improvements in anxiety (POMS), rumination, scepticism, and excitement (three circle model), which all reduced significantly. The effect size was largest for nature connection.

Subgroup analyses revealed that female adolescents’ anxiety and scepticism scores decreased substantially compared with male’s scores, which showed a subtle increase. This is consistent with Barton et al. [27], who found that wilderness expeditions were especially effective for females in improving self-esteem and nature connection. Those with low nature connection scores at baseline increased in nature connection substantially, whereas those who scored highly in nature connection at baseline showed no change in nature connection scores by post-intervention.

These survey results showing improved scores in nature connection, anxiety, and social connection, are consistent with previous research, which found that nature-based interventions improved nature connection [6,21,27,39], reduced anxiety [23,26], and increased social connection, as measured through prosociality [23]. The current study also provides novel data, showing reductions in rumination and scepticism about forest bathing.

### 4.2. Interview Results

The qualitative findings indicate that, although there was a mixed degree of understanding and expectations from adolescents, the majority recognised that the key element involved being with nature. They reported an assortment of mental states before the intervention commenced, in particular, anxiety about completing schoolwork. However, following the intervention, most of the participants experiences were expressed as positive in several ways. These included being able to enjoy the activities and the social aspects of being part of a group. The participants were enabled to learn, trust, and build confidence for sharing thoughts and experiences with each other. There was also a strong indication that they were able to be present in the moment by noticing the natural world around them. They found the effects of this intervention immediately calming and relaxing.

The themes of enjoying the social aspects of being part of a group, being present in the moment by noticing the natural world, and finding the experience calming and relaxing, show some consistency with previous qualitative studies that have examined the benefits of nature for young people. For example, Birch et al. [40] found that spending time in nature offered a range of wellbeing benefits, including experiences of nature as being accepting and relational, offering a sense of connection and care with the human and non-human world. In a study where adolescents accessed nature through virtual reality, interviews revealed experiences of calm and relaxation [41].

Through focus groups [42,43] and photo surveys [44], adolescents reported that they valued nature for providing them with good times with family and friends, where they could escape day-to-day stresses, relax, and ‘unplug’. Through a combination of photovoice, ‘talking circles’, and interviews, young people revealed that they found calm, hope, and metaphors of resilience in nature [45]. When interviewed about ‘their most connected moments in nature’, the most memorable experiences involved deep sensory immersion (such as one might find in forest bathing), where they experienced calm and relaxation [46]. Something that these previous studies found which our study did not, is that young people also valued nature for the excitement, adventure, and risk it offered. In contrast, our surveys revealed a reduction in high-arousal positive affect, such as excitement, following forest bathing.

## 5. Limitations

The recruitment strategy for the study was opportunistic sampling from Scout groups in the local area. Opportunistic sampling has limitations in the respect that adolescents in a specific localised area may not be demographically or socio-economically representative of the wider population. One strength, however, is that because this was a pre-established group, a more pragmatic sample was obtained, which included adolescents who would not normally choose to engage in more mindful and creative nature connection activities. This is likely to have provided a more balanced view of how acceptable and effective the intervention was. Future studies should aim to recruit more diverse populations, and to base the sample size on a power calculation.

Gathering survey or diary data from adolescents can pose challenges, such as problems of validity and reliability, associated with inaccurate assessment, recall bias, and social desirability bias [47]. Some adolescents did not wish to complete surveys because it was viewed as too similar to performing schoolwork, this is reflected in the lower numbers of survey completion (43% loss) compared with numbers originally registered for the session. Future research could find more engaging methods of evaluation that appeal to adolescents’ creativity. For example, author KM has been piloting a more creative version of the three circle model of affect measure, where adolescents are invited to vote using different coloured leaves or different objects (e.g., sticks, leaves, and stones) to represent how they feel in relation to the three circle model of affect (i.e., excited, calm, or anxious) at the start and end of the session. Items are placed anonymously in a bag and the guide then transfers these to a drawing of the three circles, and photographs and counts the number of items corresponding to each emotion (an example can be found in the Appendix A).

The researchers observed that some adolescents appeared self-conscious about providing interview responses. In a paper outlining recommendations for interviewing young people, Ponizovsky-Bergelson et al. [48] suggest that interviews that include encouragement, open-ended questions, or question requests (e.g., ‘can you tell me about your artwork?’) produce the richest data.

This study did attempt to collect heart rate variability (HRV) data. However, adolescents were too mobile and continuously adjusted their HRV devices (Polar H10), meaning that devices frequently failed to provide a signal. Hence, data were only successfully gathered for seven participants, and we do not report this data here as the sample size is too small to be generalisable. 

Finally, there is great variability in delivering nature-based programmes, and this makes it a challenge to objectively compare the current intervention with interventions assessed in previous studies [49]. Future research would benefit from the inclusion of a direct control group, which could take the form of mere access to nature vs. mere active nature connection interventions.

## 6. Conclusions

There is a wealth of evidence showing that mere access to nature can benefit adolescents’ wellbeing. However, nature connection interventions that actively seek to develop a connection with nature can have an even greater impact on adolescent wellbeing, and are currently less researched. The current study assessed an active nature connection intervention, and found that following urban forest bathing, significant improvements were found in anxiety, rumination, scepticism, nature connection, and social connection. The largest effect size (improvement) was found for nature connection. Interview data complemented the survey data, and provided convergent validity. Interviews revealed that participants enjoyed the social aspects of being part of a group, being mindfully present in the moment by noticing the natural world around them, and feeling immediately calmed and relaxed. Previous nature connection intervention studies have assessed interventions taking place in wilderness or rural settings. However, some researchers suggest that young people might be limited in their access to such settings due to their lack of autonomy, transport, and parental permission [4]. The current study took place in a busy urban park in a capital city (London, UK), and demonstrates significant improvements in nature connection and wellbeing measures, which were echoed by complementary interview data. Urban forest bathing improved nature connection and wellbeing in adolescents, and could be implemented and/or signposted by schools, youth groups, and charities, such as the Scouts and National Youth Agency, to improve nature connection and wellbeing in young people. Future research would benefit from the inclusion of a control group, perhaps comparing more passive nature access (for which there is more research) to more active nature connection interventions (for which there is less research), to assess whether one is superior to the other.

## Figures and Tables

**Figure 1 ijerph-19-15594-f001:**
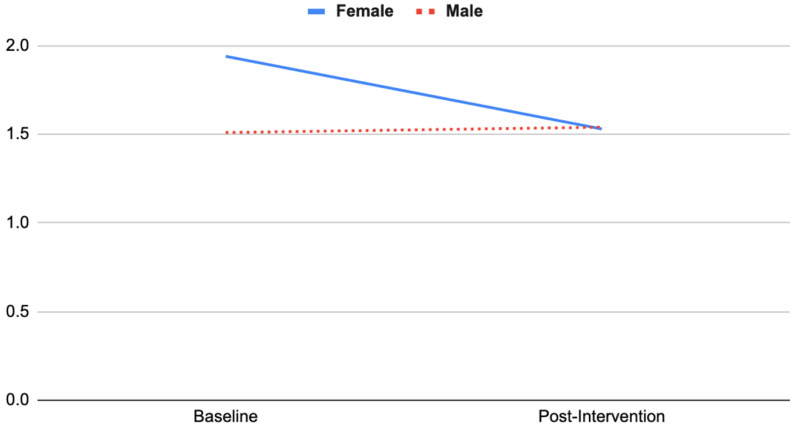
Interaction effect between gender and anxiety.

**Figure 2 ijerph-19-15594-f002:**
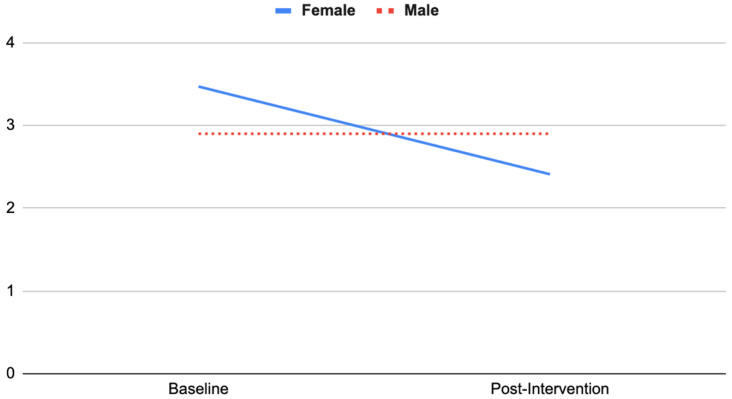
Interaction effect between gender and scepticism.

**Figure 3 ijerph-19-15594-f003:**
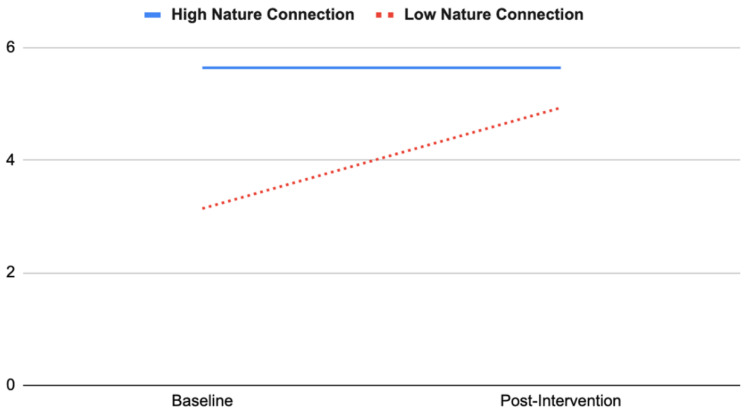
Interaction effect between baseline nature connection and change in nature connection.

**Table 1 ijerph-19-15594-t001:** Changes in outcome variables between baseline and post-intervention.

	Pre-Intervention		Post-Intervention			
	*M*	*SD*	*M*	*SD*	*t*	*d*
Nature Connection ***	3.85	1.42	5.13	1.24	5.90	0.95
Anxiety (POMS) *	1.75	0.63	1.55	0.60	−2.08	−0.33
Rumination **	2.92	1.53	2.03	1.27	−2.99	−0.48
Scepticism *	3.10	1.79	2.64	1.83	−2.07	−0.33
Excitement *	1.13	0.34	1.03	0.16	−2.08	−0.34
Calmness	1.77	0.43	1.82	0.39	0.57	0.09
Anxiety	1.67	0.93	1.67	1.06	0.00	0.00
Social Connection **	4.00	1.12	4.56	1.10	2.91	0.47

* *p* < 0.05, ** *p* < 0.01, *** *p* < 0.001.

## Data Availability

Anonymised data are available on request by contacting the corresponding author.

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
