# Peer review of "‘This Is What the Colour Green Smells Like!’: Urban Forest Bathing Improved Adolescent Nature Connection and Wellbeing"

_ijerph, 2022, doi:10.3390/ijerph192315594_

Round 1

Reviewer 1 Report

I was pleased to read the manuscript entitled "‘This is what the colour green smells like!’: Urban Forest Bath-2 ing improved adolescent nature connection and wellbeing" and to review it.

The study was focused on showing that mere access to nature can benefit adolescents’ wellbeing. The article is interesting to read because it reveals the various reactions of teenagers to green nature. However, from a scientific point of view, the paper seems a little weak.

1. There is no clearly defined research aim and objectives. It is common to state the aim of the study after an overview of the main research in the scientific field and presented it at the end of the introduction section. The study aim should also be presented in the abstract. It is also unclear what scientific hypotheses the authors want to test in this study.

2. In the introduction, the authors presented many studies that present the importance of being in nature for the well-being of adolescents. This study is of a similar nature. Therefore, it is important to emphasize the novelty of this study, new aspects and results of the study.

Thank you for considering my opinion. I encourage authors to keep on working to improve the manuscript.

Author Response

Reviewer 1

I was pleased to read the manuscript entitled "‘This is what the colour green smells like!’: Urban Forest Bathing improved adolescent nature connection and wellbeing" and to review it.

The study was focused on showing that mere access to nature can benefit adolescents’ wellbeing.

Authors response: The current study is based on connection to nature via intervention, rather than mere access to nature. We have removed a paragraph from the introduction so that these distinct literature reviews are now side by side and can be more easily comparable for readers.

  1. There is no clearly defined research aim and objectives. It is common to state the aim of the study after an overview of the main research in the scientific field and presented it at the end of the introduction section. The study aim should also be presented in the abstract. It is also unclear what scientific hypotheses the authors want to test in this study.

Authors response: The study aim was already present in the abstract, we have re-phrased this sentence to highlight its presence better. Apologies for the oversight in missing the aim at the end of the Introduction, we have now included the aim and hypothesis as follows:

Aim

This research aimed to assess whether an urban nature connection activity (called ParkBathe) could improve adolescents' nature connection and wellbeing, by evaluating their experiences before and after the intervention through surveys and interviews. We hypothesise that nature connection and wellbeing survey scores will improve following a formal nature connection activity which is based on Forest bathing.

  1. In the introduction, the authors presented many studies that present the importance of being in nature for the well-being of adolescents. This study is of a similar nature. Therefore, it is important to emphasize the novelty of this study, new aspects and results of the study.

Authors response: We have reorganised the paragraphs in the Introduction to make it more apparent what the unique aspects of the study are: 1) delivery of a nature connection intervention in an experimental design rather than observation of mere access to nature in a correlational design; and 2) urban rather than wilderness/rural setting to provide accessibility to young people. This paragraph appears before the Aims and reads as follows:

 These formal nature connection intervention studies provide promising evidence for the potential to improve adolescent nature connection and wellbeing. However, most previous studies have assessed mere access to nature, relying on correlational survey data, and have not used experimental designs which involve assessing outcomes before and after delivering a more active nature connection intervention. The current study is therefore unique in using mixed-methods, experimental design collecting survey and interview data, before and after a formal nature connection intervention. Whilst most previous intervention studies conducted Forest bathing interventions in wilderness or rural settings, young people might be limited in their access to such wilderness or rural settings due to their lack of autonomy, availability of transport and parental permission (Skar et al. 2016). The current study is unique in the respect that the intervention is situated in more accessible urban parks and encourages adolescents to connect with nearby nature, with the goals of increasing feasibility of access and ease of continued practice.

Reviewer 2 Report

In abstract include sampling method, sample size, methodology and statistical tests. linee156-156 can be placed at end of participants sect. to clarify subject numbers and flow

add more detail to sampling procedure

limitations sect. address sampling strategy limitations

also there is great variability in delivering nature-based programs

overall a good study. Mixed methods helps with convergent validity

Can be improved by following JARS, CONSORT or PRISMA reporting guidelines

Author Response

Reviewer 2

In abstract include sampling method, sample size, methodology and statistical tests. linee156-156 can be placed at end of participants sect. to clarify subject numbers and flow

Authors response: Thank you for these helpful and practical comments. We have moved the line as suggested and added the details you suggested to the Abstract as follows:

Method: In an experimental repeated measures design, 44 adolescents sampled opportunistically from Scouts groups, completed surveys and interviews before and after experiencing an urban nature connection intervention. Results: Paired-samples t tests between baseline and post-intervention survey scores revealed….

add more detail to sampling procedure

Authors Response:  We have added the following details to the Methods section regarding the sampling procedure:

Adolescents were sampled opportunistically from Scouts groups. The authors contacted Scout Leaders local to the intervention site and asked them to invite their Scout groups and obtain parental permissions and consent. Scout Leaders attended the sessions with their Scout Groups to provide a sense of safeness through familiarity and also to comply with safeguarding requirements.

limitations sect. address sampling strategy limitations

Author response: We have added the following text to the Limitations section concerning sampling limitations:

The recruitment strategy for the study was opportunistic sampling from Scout groups in the local area. Opportunistic sampling offers limitations in the respect that adolescents in a specific localised area may not be demographically or socio-economically representative of the wider population. One strength, however, is that because this was a pre-established group, a more pragmatic sample was obtained which included adolescents who would not normally choose to engage in more mindful and creative nature connection activities. This is likely to have provided a more balanced view of how acceptable and effective the intervention was. Future studies should aim to recruit more diverse populations and to base the sample size on a power calculation.

also there is great variability in delivering nature-based programs

Authors response: Good point, we have added the following to the limitations section:

Finally, there is great variability in delivering nature-based programs and this makes it a challenge to objectively compare the current intervention with interventions assessed in previous studies. Future research would benefit from the inclusion of a direct control group, which could take the form of mere access to nature vs mere active nature connection interventions.

overall a good study. Mixed methods helps with convergent validity

Can be improved by following JARS, CONSORT or PRISMA reporting guidelines

Authors response: Thank you for this suggestion. We have examined the Consort reporting guidelines and have included additional details in the manuscript where appropriate given the design of the study (i.e. no control group and hence non-randomised).

Reviewer 3 Report

It is a well-written manuscript and an interesting topic. The cited literature was published in recent years and is relevant to the topic. The methodology was presented clearly, and the following parts could be improved:

1. Forest bathing is an important concept and term in this research. It would be better if the author(s) could explain it in more detail. For example, the author(s) could mention what experience participants had in terms of forest bathing. What did they see? What did they experience? What were the high nature connection and low nature connection?

2. The conclusion section is too short. The author(s) should justify the theoretical and practical contributions of this research.

Good luck with your revision!

Author Response

Reviewer 3

It is a well-written manuscript and an interesting topic. The cited literature was published in recent years and is relevant to the topic. The methodology was presented clearly, and the following parts could be improved:

  1. Forest bathing is an important concept and term in this research. It would be better if the author(s) could explain it in more detail. For example, the author(s) could mention what experience participants had in terms of forest bathing. What did they see? What did they experience? What were the high nature connection and low nature connection?

Authors response: Details of the intervention can be found in the methods section (subheading Intervention). We have now referred to this in the Introduction as follows:

In this study Forest bathing was delivered to adolescents and so accordingly was adapted to suit the needs of younger people by encompassing more playful and creative sensory activities, such as creating artwork from natural materials and interviewing a tree about its life (more details can be found in the methods section).

We have also added more detail about interventions from previous studies in the Introduction as follows:

Most of the nature connection activities described in previous literature appear to be Forest School or Forest bathing inspired. According to the Forest School Association, Forest School is a child-centred learning process which supports play, exploration and risk taking through practical experiences in a natural setting. Forest bathing is a slow mindful nature walk where participants are guided to pay close attention to their surroundings using all of their senses. Where applied to children, Forest bathing appears to take more diverse forms and might encompass playful and creative sensory activities.

  1. The conclusion section is too short. The author(s) should justify the theoretical and practical contributions of this research.

Authors response: Thank you for this suggestion. We have included more detail in the conclusion section and related the findings more directly to the unique aspects of the study and the aims and hypotheses the study strove to address as follows:

There is a wealth of evidence showing that mere access to nature can benefit adolescents’ wellbeing. However, nature connection interventions which actively seek to develop a connection with nature can have an even greater impact on adolescent wellbeing and are currently less researched. The current study assessed an active nature connection intervention and found that following urban Forest bathing, significant improvements were found in anxiety, rumination, scepticism, nature connection, and social connection. The largest effect size (improvement) was found for nature connection. Interviews data complemented the survey data and provided convergent validity. Interviews revealed that participants enjoyed the social aspects of being part of a group, being mindfully present in the moment by noticing the natural world around them and feeling immediately calmed and relaxed. Previous nature connection intervention studies have assessed interventions taking place in wilderness or rural settings. However, some researchers suggest that young people might be limited in their access to such settings due to their lack of autonomy, transport and parental permission (Skar et al. 2016).  The current study took place in a busy urban park in a capital city (London, UK) and demonstrated significant improvements in nature connection and wellbeing measures which were echoed by complementary interview data. Urban Forest bathing improved nature connection and wellbeing in adolescents and could be implemented or signposted to, by Schools and Youth groups and charities such as the Scouts and National Youth Agency to improve nature connection and wellbeing in young people. Future research would benefit from the inclusion of a control group, perhaps comparing more passive nature access (for which there is more research) to more active nature connection interventions (for which there is less research), to assess whether one is superior to the other.

Round 2

Reviewer 1 Report

I found that the authors of the article responded in detail to my comments and those of the other reviewers.